

# Remote collection of microorganisms at two depths in a freshwater lake using an unmanned surface vehicle (USV)

Craig Powers[1], Regina Hanlon[2] and David G. Schmale III[2]

[1] Department of Civil and Environmental Engineering, Virginia Polytechnic Institute and State University (Virginia Tech), Blacksburg, VA, United States of America
[2] Department of Plant Pathology, Physiology, and Weed Science, Virginia Polytechnic Institute and State University (Virginia Tech), Blacksburg, VA, United States of America

## ABSTRACT

Microorganisms are ubiquitous in freshwater aquatic environments, but little is known about their abundance, diversity, and transport. We designed and deployed a remote-operated water-sampling system onboard an unmanned surface vehicle (USV, a remote-controlled boat) to collect and characterize microbes in a freshwater lake in Virginia, USA. The USV collected water samples simultaneously at 5 and 50 cm below the surface of the water at three separate locations over three days in October, 2016. These samples were plated on a non-selective medium (TSA) and on a medium selective for the genus *Pseudomonas* (KBC) to estimate concentrations of culturable bacteria in the lake. Mean concentrations ranged from 134 to 407 CFU/mL for microbes cultured on TSA, and from 2 to 8 CFU/mL for microbes cultured on KBC. There was a significant difference in the concentration of microbes cultured on KBC across three sampling locations in the lake ($P = 0.027$), suggesting an uneven distribution of *Pseudomonas* across the locations sampled. There was also a significant difference in concentrations of microbes cultured on TSA across the three sampling days ($P = 0.038$), demonstrating daily fluctuations in concentrations of culturable bacteria. There was no significant difference in concentrations of microbes cultured on TSA ($P = 0.707$) and KBC ($P = 0.641$) across the two depths sampled, suggesting microorganisms were well-mixed between 5 and 50 cm below the surface of the water. About 1 percent (7/720) of the colonies recovered across all four sampling missions were ice nucleation active (ice+) at temperatures warmer than $-10\,°C$. Our work extends traditional manned observations of aquatic environments to unmanned systems, and highlights the potential for USVs to understand the distribution and diversity of microbes within and above freshwater aquatic environments.

Corresponding author
David G. Schmale III,
dschmale@vt.edu

## INTRODUCTION

The ecology of freshwater ecosystems is linked to the temporal and spatial dynamics of aquatic microorganisms (*Beisner et al., 2006*). Microorganisms play an important role in the food web as drivers and indicators of ecosystem health (*Newton et al., 2011*; *Shafi et al., 2017*). Biological ice nucleators such as *Pseudomonas syringae* have been collected

throughout the water cycle, and have been implicated as drivers of precipitation processes (*Ichinose, Taguchi & Mukaihara, 2013*; *Morris et al., 2008*). New research is needed to understand and predict the abundance, distribution, and diversity of microorganisms in freshwater lakes (*Morris et al., 2008*). *Humayoun, Bano & Hollibaugh (2003)* observed differences in microbial diversity at different depths in Mono Lake, California, USA. *Song et al. (2007)* examined distributions of toxins from algal blooms along the water column in Lake Taihu, Wuxi, China. *Pietsch, Vinatzer & Schmale III (2017)* reported high concentrations of *P. syringae* in Claytor Lake, VA, USA from the surface down to almost 10 m. These authors also showed that concentrations of *P. syringae* varied dramatically from day to day and location to location (*Pietsch, Vinatzer & Schmale III, 2017*). Though these observations have provided important data on the distribution of microorganisms in aquatic environments, they have been limited by manned collections (i.e., at least one human was needed to collect the samples). In this study, we extend these manned observations to unmanned systems, highlighting the potential for robots to collect samples to study the distribution and diversity of microbes within and above freshwater aquatic environments.

Recent advances in unmanned systems have created new possibilities for sampling natural and managed ecosystems (*Pennington, Blum & Chavez, 2016*). Though these unmanned systems are seeing increased use in a variety of scientific applications, many challenges must be overcome. First, long-range control and communication of unmanned systems require high data rates, often more than satellite-based communication can provide. Commercial Wi-Fi systems are one potential solution to fill this need (*Takahata et al., 2016*). Such systems have been used for surface communication in autonomous underwater vehicles with a range of 1 km (*Stokey et al., 2005*), and in specialized point to point sustained networks at ranges of over 100 km (*Flickenger et al., 2008*). Second, object avoidance is an extremely important aspect of unmanned operations and becomes an absolute requirement when operating in existing complex manned boat traffic. Adaptive evaluation schemes and algorithms have been recently developed and tested to allow autonomous watercraft to meet international regulations for preventing collisions at sea (*Kuwata et al., 2014*; *Shah et al., 2016*).

In this manuscript, we describe the development and deployment of a USV to sample water remotely at multiple depths and locations in a freshwater lake. This integrated system was used to test the hypothesis that concentrations of microorganisms in Claytor Lake, VA vary with depth, geographic location, and date of sampling. The specific objectives of this work were to (1) develop an automated water sampler to remotely collect samples of water with a USV in a freshwater lake, (2) conduct a series of field experiments to remotely collect samples of water with the USV at two depths at three different locations in the lake, and (3) culture microorganisms from the water samples on a non-selective medium (TSA) and on a medium selective for the genus *Pseudomonas* (KBC) to estimate concentrations of culturable bacteria in the lake. Our work represents a unique approach to collect and characterize the distribution of microorganisms in aquatic environments, and could be extended to tracking the movement of hazardous agents during algal blooms and in floodwaters generated by hurricanes and other extreme weather events.

## MATERIALS AND METHODS

### Study site and design

Samples were collected over three consecutive days on 25, 26, and 27 October, 2016 in Claytor Lake, VA, USA. This freshwater lake has an approximate surface area of 18.2 square kilometers. Samples were collected in a cove located at N37°2.34′4″W80°37.7′9″. The cove was selected in part due to its isolation; no manned boats (other than our pontoon boat used as our base station) visited the cove before or during our sampling operations. Water samples were collected at two depths (5 and 50 cm) below the water surface for three locations, L1 (N37°2.33′4″W80°37.06′6″), L2 (N37°2.30′5″W80°37.07′7″), and L3 (N37°2.35′9″W80°37.12′4″) (Fig. 1). Depths of the lake at each of the sampling locations were measured with a portable sonar sensor (Signstek FF-003, Wilmington, DE, USA). The sensor was placed 20 cm below the surface of the water, and the depth reader output screen was monitored until the continuous reading on the screen did not vary more than 0.5 m for 10 readings. Depths for each of the locations were measured at 11 m for L1, 10 m for L2, and 4 m for L3. Sampling dates, times, and locations are provided in Table 1. Each location was separated by the transition time of the USV between each location. GPS coordinates were recorded for all navigation paths and each sample location (Fig. 1). Though water temperature is an environmental variable, it was assumed to be constant across the distances and times within and among sample collections in this study.

### Unmanned surface vehicle (USV)

A Clearpath Robotics Kingfisher USV (Clearpath Robotics, Kitchener, Ontario, Canada) (Fig. 2) was used as our water-sampling platform. This electric USV weighed 28 kg, was 1.35 m × 0.98 m × 0.32 m ($L \times W \times H$), had a maximum payload of 10 kg, and was operated via remote control (the vehicle is equipped with autonomous navigation through a ground control station, but this was not used during our mission because of limitations with obtaining consistent GPS signals within the cove of interest). Each removable 29 Ah battery provided up to 3 h of continuous runtime. Differential thrust provided steering via the two impeller thrusters and allowed for a maximum speed of 1.7 m/s. This platform enabled extremely precise, controlled movements in shallow waters and also minimized water disturbances to the surrounding aquatic environment that could be caused by the vehicle.

### Development of an automated water sampler onboard a USV

A water sampler was developed for deployment on a USV with the goal of sampling at two distinct depths for three locations (Fig. 3). A vacuum system and flow rate was used to maximize the water collection rate while minimizing turbulent fluid flow at the collection nozzle. Seven 1 L bottles were used as the collection vessels. Two were used at each sampling location, with each of the bottles holding a sample from each of the two depths (5 and 50 cm). One bottle was used as a control (no sample). Each bottle was attached via Tygon B-44-4X PVC 1/4″ ID, 3/8″ OD tubing to an air pump (ZT370-01, Dongguan Zhentian precision electronic co., LTD, Dongguan, Guangdong, China) calibrated to pull approximately 0.5 Lpm. Each bottle had a section of this tubing attached to a stainless

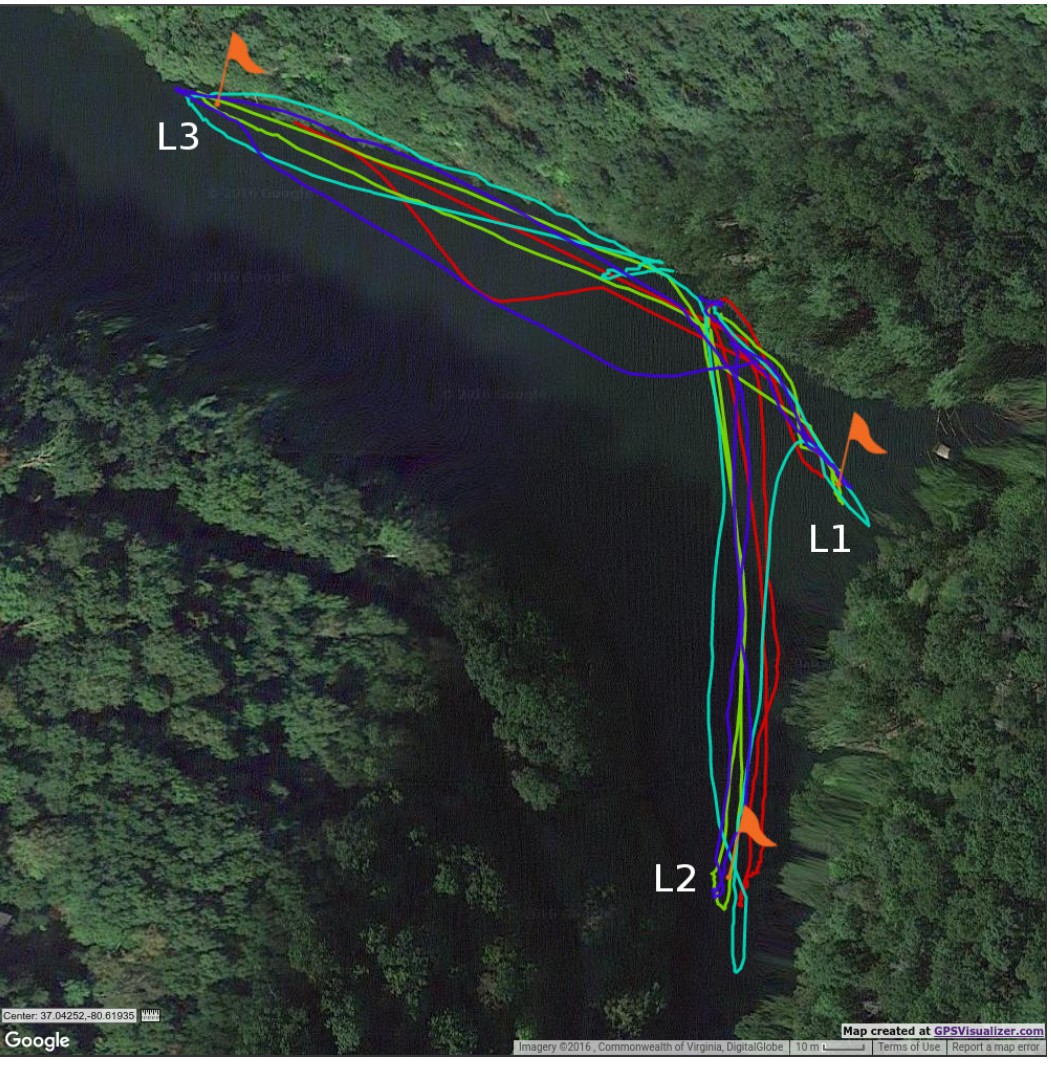

**Figure 1** **Image of sampling routes and locations.** Unmanned surface vehicle (USV) navigation routes (Mission 1, red; Mission 2, green; Mission 3, cyan; and Mission 4, blue) and sample locations L1, L2, and L3 (orange flags) for the four missions conducted from October 25th to October 27th. The USV collected 500 mL of water from at two depths (5 cm and 50 cm) for each location. Map data ©: Google 2016 Commonwealth of Virginia, DigitalGlobe.

steel tube that served to siphon the water directly from the lake. Six changeable siphons (one for each depth at each of three sampling locations) were attached to arms on the water sampler platform via 3D printed bevel gears (Supplemental Information) that where actuated by waterproof servos (SX401WP, Hobbico Inc., Champaign, IL, USA). Two lengths of stainless steel tube were chosen to allow for simultaneous collections at depths of 5 and 50 cm for each location. The tubing ends were bent into slight curves so that a water trap existed while the tubes were in the horizontal, stowed position (Fig. 2). Aliquots of 10 mL of sterilized water were placed into each of the six tubing water traps to act as an environmental seal to prevent air flow movement and contamination of the sampling tube

**Table 1  Microbes cultured from USV samples for different media types.** Mean concentrations of bacteria (CFU/mL) for KBC and TSA media. Data are reported for four sampling missions, over three days, at three locations at Claytor Lake, VA (L1, L2, and L3). Remote collections were performed with a water sampler onboard an unmanned surface vehicle (USV), and ~500 mL of water was collected for each depth (5 and 50 cm) and location.

| Mission | Date | Time | Location | Depth | Mean CFU/mL KBC media ± SD | Mean CFU/mL TSA media ± SD |
|---|---|---|---|---|---|---|
| 1 | 25-Oct-16 | 11:00 | L1 (37.0426, −80.6185) | 5 cm | 6.3 ± 0.95 | 196 ± 53.7 |
| 1 | 25-Oct-16 | 11:00 | L1 (37.0426, −80.6185) | 50 cm | 5.3 ± 2.10 | 324 ± 22.6 |
| 2 | 25-Oct-16 | 11:05 | L2 (37.0418, −80.6188) | 5 cm | 4.6 ± 0.60 | 336 ± 36.8 |
| 2 | 25-Oct-16 | 11:05 | L2 (37.0418, −80.6188) | 50 cm | 4.8 ± 0.69 | 407 ± 86.3 |
| 3 | 25-Oct-16 | 11:10 | L3 (37.0433, −80.6201) | 5 cm | 3.7 ± 0.61 | 291 ± 24.0 |
| 3 | 25-Oct-16 | 11:10 | L3 (37.0433, −80.6201) | 50 cm | 2.0 ± 0.69 | 251 ± 80.6 |
| 4 | 26-Oct-16 | 11:00 | L1 (37.0426, −80.6185) | 5 cm | 2.8 ± 0.40 | 198 ± 19.8 |
| 4 | 26-Oct-16 | 11:00 | L1 (37.0426, −80.6185) | 50 cm | 2.6 ± 0.20 | 134 ± 5.66 |
| 5 | 26-Oct-16 | 11:05 | L2 (37.0418, −80.6188) | 5 cm | 3.5 ± 0.95 | 196 ± 17.0 |
| 5 | 26-Oct-16 | 11:05 | L2 (37.0418, −80.6188) | 50 cm | 2.7 ± 0.50 | 252 ± 79.2 |
| 6 | 26-Oct-16 | 11:10 | L3 (37.0433, −80.6201) | 5 cm | 4.1 ± 0.90 | 148 ± 31.1 |
| 6 | 26-Oct-16 | 11:10 | L3 (37.0433, −80.6201) | 50 cm | 3.7 ± 1.17 | 245 ± 1.41 |
| 7 | 27-Oct-16 | 11:00 | L1 (37.0426, −80.6185) | 5 cm | 4.5 ± 1.10 | 390 ± 73.5 |
| 7 | 27-Oct-16 | 11:00 | L1 (37.0426, −80.6185) | 50 cm | 8.1 ± 1.45 | 373 ± 32.5 |
| 8 | 27-Oct-16 | 11:05 | L2 (37.0418, −80.6188) | 5 cm | 5.0 ± 0.35 | 342 ± 5.66 |
| 8 | 27-Oct-16 | 11:05 | L2 (37.0418, −80.6188) | 50 cm | 5.0 ± 1.11 | 321 ± 157 |
| 9 | 27-Oct-16 | 11:10 | L3 (37.0433, −80.6201) | 5 cm | 3.8 ± 0.92 | 363 ± 46.7 |
| 9 | 27-Oct-16 | 11:10 | L3 (37.0433, −80.6201) | 50 cm | 3.9 ± 1.10 | 340 ± 19.8 |
| 10 | 27-Oct-16 | 13:00 | L1 (37.0426, −80.6185) | 5 cm | 7.3 ± 1.75 | 311 ± 123 |
| 10 | 27-Oct-16 | 13:00 | L1 (37.0426, −80.6185) | 50 cm | 5.8 ± 2.51 | 256 ± 127 |
| 11 | 27-Oct-16 | 13:05 | L2 (37.0418, −80.6188) | 5 cm | 6.4 ± 0.60 | 384 ± 33.9 |
| 11 | 27-Oct-16 | 13:05 | L2 (37.0418, −80.6188) | 50 cm | 7.2 ± 0.69 | 321 ± 35.4 |
| 12 | 27-Oct-16 | 13:10 | L3 (37.0433, −80.6201) | 5 cm | 2.6 ± 1.97 | 227 ± 24.0 |
| 12 | 27-Oct-16 | 13:10 | L3 (37.0433, −80.6201) | 50 cm | 5.4 ± 1.56 | 358 ± 19.8 |

while the USV proceeded through the sampling cycle. At the start of the collection cycle at each location, one sampler arm transitioned from a rear horizontal stowed position into the water, then past the vertical position and to a forward horizontal position, 175 degrees from the starting position, in order for the sterilized water to flow out of the tubing. This cleared the line of sterile water preparing the system for water collection. The arm then transitioned back into the water to a vertical position. Approximately 500 mL of water was then siphoned simultaneously for both depths into a collection bottle for each depth. The arm then transitioned back to the horizontal stowed position where water in the tubing (leftover from the collection process) would stay in the water trap at the end of the tubing. This water served to prevent contamination from the air while the USV completed the collection cycle. The stainless steel tubing was disinfested with ethanol and flushed with sterile water after each sampling day, and on the last day sterile water was used to flush the tubing between the two sampling periods for that day. A microcontroller (keyestudio MEGA 2560, Shenzhen, China) in conjunction with servo and motor driver circuits, were

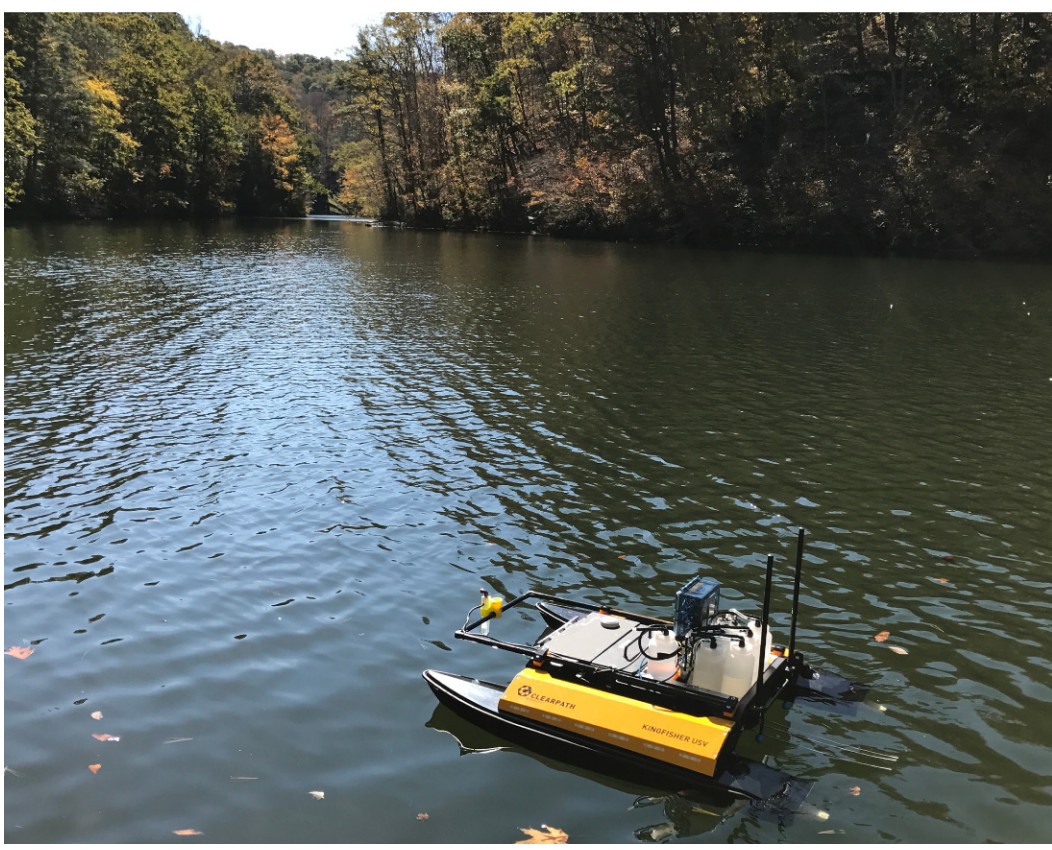

**Figure 2** **Image of USV on lake.** The Clearpath Robotics Kingfisher USV equipped with the automated water sampler on Claytor Lake, VA, USA. The water sampler onboard the USV collected samples of water at two different depths. Each sample was collected into sterile 1 L bottles. The operation of the vacuum pumps and sampling tubes were remote-controlled through a microcontroller and supporting electronics housed in a waterproof box. A collection bottle containing 500 mL of sterile water was used as an unexposed control during each mission. Image courtesy of D. Schmale.

integrated into the water sampler and USV to control motion and timing of the servos and pumps. The microcontroller was then controlled through the computer on the USV via a wireless link to a portable computer running a Linux operating system.

## Processing of samples for culturable bacteria

After the return of the USV from each collection cycle, the lake water samples were removed from the USV and capped with sterilized screw cap lids. After all collections for the day were completed, samples were transported on ice in a cooler to the lab for processing in the same 1 L collection bottle (Nalgene #2187-0032, Thermo Fisher Scientific, Waltham, MA, USA). Subsamples of 250 mL from each collection bottle were filtered through a 47 mm diameter, 0.22 mm GSWP nitrocellulose filter (#9004-70-0, Millipore, Burlington, MA, USA) in a sterile holder and receiver unit (Nalgene #300-4100, Thermo Fisher Scientific, Waltham, MA, USA). Filters were transferred to sterile 100 mL bottles with 5 mL of the filtrate and a stir bar. Samples were stirred for 10 min to resuspend microorganisms from

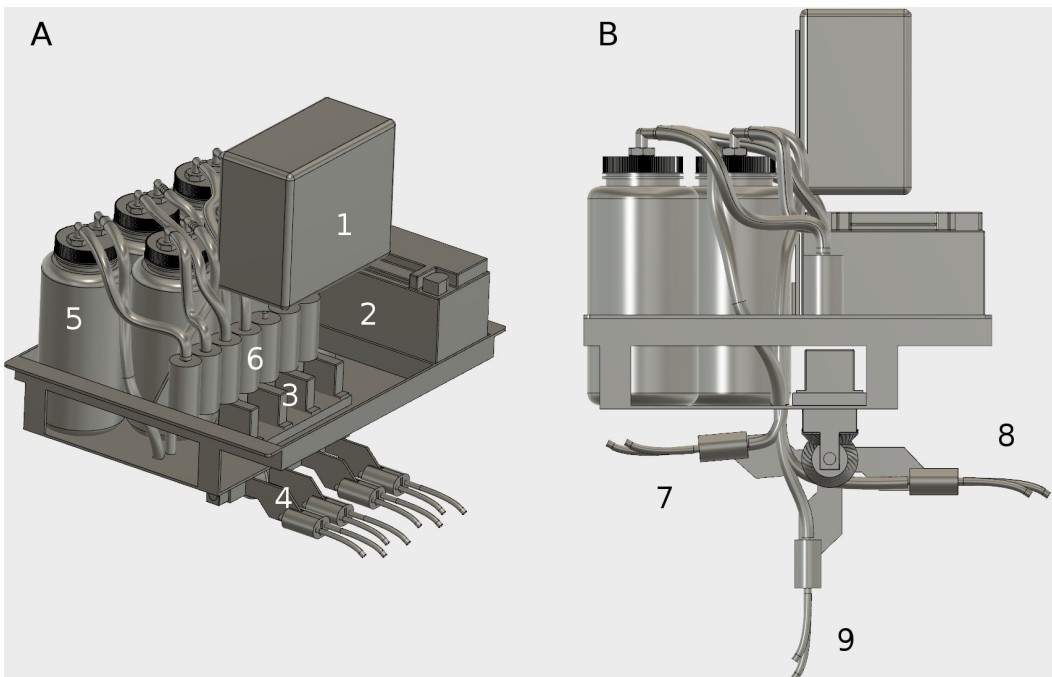

**Figure 3 Engineering design of the automated water sampler onboard the USV.** The sampler (A) consisted of a microcontroller and electronics (1) powered by a lithium iron phosphate battery (2) that actuated servos (3) to control siphon positions (4) and pump water into 1 L containers (5) with vacuum pumps (6). During vehicle movement, the siphon arms (B) were in the stowed configuration (7). Upon reaching the sampling location, the appropriate arm rotated 175 degrees (8) to allow the sterile water in the tube to flow out. The arm then rotated back 90 degrees (9) into the water for collection. After a set amount of collection time, the arm returned to the stowed configuration (7) before proceeding to the next location.

the filter surface (resulting in a 50× concentration). A 0.5 dilution of this resuspension (25× concentration) was used for plating. Kings B medium plus cycloheximide (KBC), a medium semi-selective for *Pseudomonas* (*Mohan & Schaad, 1987*), was used to plate 200 µL of both the 25× and 50× concentrations in triplicate. This medium consisted of 15 g/L proteose peptone, 1.5 g/L anhydrous $K_2HPO_4$, 10 mL/L 100% glycerol, 6 mM $MgSO_4$ with 24 mM $H_3BO_3$, cephalexin (10 mg/L), and cycloheximide (50 mg/L). Resuspended filtrates were also plated onto 10% tryptic soy agar (TSA) plus cycloheximide (50 mg/L) at 25×, 5×, and 0.5× concentrations to obtain counts of culturable bacteria on TSA (*Hanlon et al., 2017*). Colonies were considered culturable if they showed visible growth on TSA after 4–6 days at 22 °C. Following incubation, colony forming units (CFUs) were recorded from each plate (entire plates were counted for KBC, and $\frac{1}{2}$ of each plate was counted for TSA), and the plates were held at 4 °C for ice nucleation assays (*Pietsch, Vinatzer & Schmale III, 2017*). The mean colony count for KBC plates at 25× concentration was 23 CFUs per plate, providing more than the minimum 10 colonies needed to screen 30 from each location from three plate replicates. This corresponded to 5 mL of lake sample, and a mean of 4.6 CFU/mL. For TSA plates, the mean colony count at 5× concentration was 290 CFUs per plate. This corresponded to 1 mL of lake sample, and a mean of 290 CFU/mL.

### Ice nucleation assays

For ice nucleation assays, ten colonies were taken from three replicates of KBC plates, for a total of 30 colonies from each sampling location. A total of 720 colonies was selected at random (180 from each of the four missions). Each colony was picked with a sterile toothpick and transferred to 140 μL of water. Two droplets of 12 μL of each sample were pipetted onto floating boats made of Parafilm® M placed on top of a cooling bath (Lauda Alpha RA 12, LAUDA-Brinkmann, LP, Delran, NJ, USA) (*Hanlon et al., 2017*). Droplets of sterile water were used as negative controls. Samples were loaded onto the floating boats at −5 °C, and the temperature of the bath was then lowered to −12 °C in one degree increments. Freezing temperatures were recorded for all of the frozen droplets. Microbes from droplets freezing at temperatures warmer than −10 °C were considered to be ice nucleation active (designated as ice +), and were identified using portions of 16S rDNA sequences as described by *Pietsch, Vinatzer & Schmale III (2017)*.

### Statistical analyses

Statistical analyses were conducted using R version 3.4.0. A hierarchical model and linear regression model were used to examine differences among culturable bacteria collected at 5 and 50 cm at three locations over three days. Four collection missions were conducted with one mission per day on days one and two, and two missions on day three. Statistical comparisons were made for missions one, two, and three (except when discussed otherwise). A Shapiro–Wilk normality test was used to verify data were approximately normally distributed and graphically checked using density plots and histograms. A 95% confidence interval was used for significant differences ($P < 0.05$).

## RESULTS

### Missions

We conducted a total of 12 sampling missions (Table 1) with an automated water sampler onboard a USV system. All missions were conducted from a pontoon boat (the mothership) anchored in place along the shore of the cove at the center of the sampling operations. Three sampling missions were completed in one continuous cycle each day for three days with an additional cycle being added on the third and last day (Table 1). During each sampling mission, the USV was controlled by remote control (RC), by the pilot in command (PIC, Schmale). The sampler was operated by a wireless link from a ground control station by a sensor operator (SO, Powers). Weather was clear with low to moderate winds and an average temperature of 11.7 °C (Mission 1), 11.9 °C (Mission 2), 12.2 °C (Mission 3), and 17.9 °C (Mission 4) during time of operation for each sampling day respectively.

### Concentrations of bacteria on KBC and TSA

Concentrations (CFU/mL) of bacteria in water sampled from Claytor Lake were calculated using colony counts from growth on KBC and TSA media (Table 1, Supplemental Information 1). Concentrations ranged from 126 CFU/mL to 468 CFU/mL for microbes cultured on TSA, and from 2 CFU/mL to 46 CFU/mL for microbes cultured on KBC (Table 1, Fig. 4). There was a significant difference in concentrations on KBC across
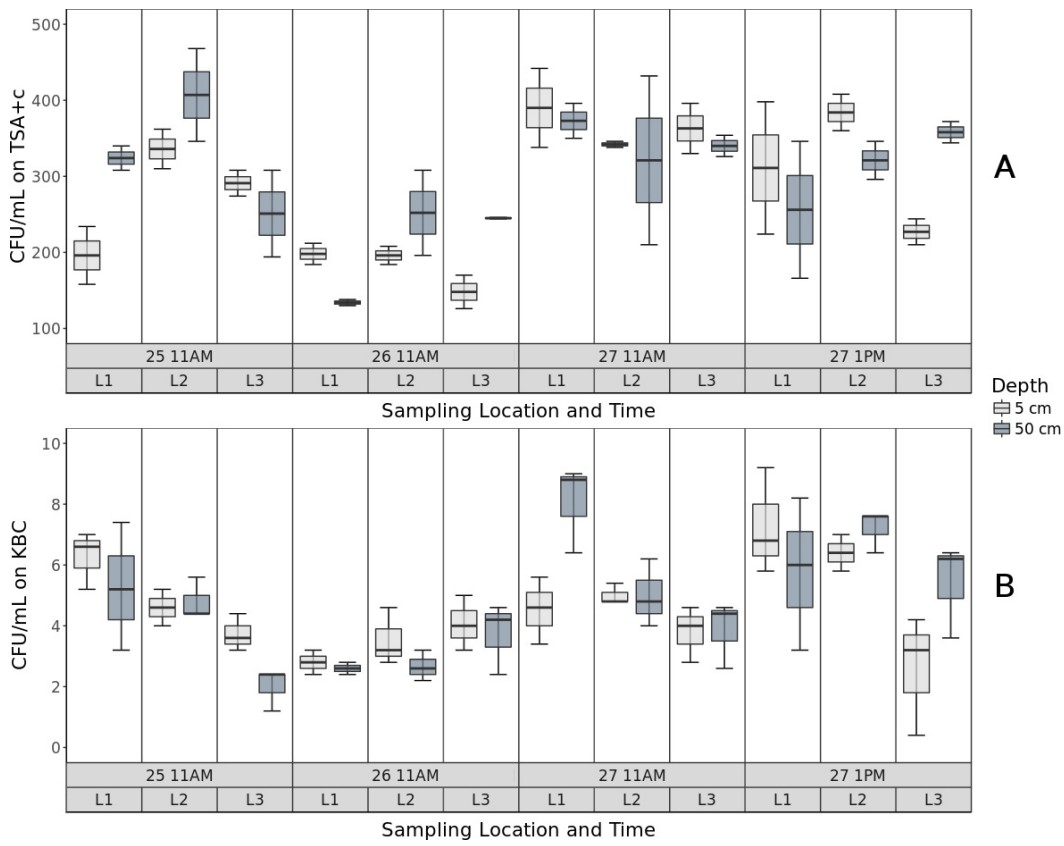

**Figure 4 CFU of bacteria on different media.** Concentrations of bacteria (CFU/mL) for TSA (A) and KBC (B) media at two depths of 5 and 50 cm at the three locations L1, L2, and L3.

three sampling locations in the lake ($P = 0.027$), suggesting an uneven distribution of *Pseudomonas* across the locations sampled. There was also a significant difference in concentrations on TSA across the three sampling days ($P = 0.038$), demonstrating daily fluctuations in concentrations of culturable bacteria. There was no significant difference in concentrations on TSA ($P = 0.707$) and KBC ($P = 0.641$) across the two depths sampled, suggesting microorganisms were well-mixed between 5 and 50 cm below the surface of the water.

## Ice nucleation assays

Seven hundred twenty colonies (180 colonies from each of the four missions) were tested for ice nucleation activity using a droplet freezing assay from $-5\,°C$ to $-12\,°C$. Of these, seven of the colonies were ice+ at temperatures $>-10\,°C$ (three ice + from mission 1, one ice + from mission 2, two ice+ from mission 3, and one ice+ from mission 4). Of the seven that were ice+, two were identified as *Pseudomonas* and one as *Xanthomonas*; genera that have been previously described as containing species of biological ice nucleators (*Hanlon et al., 2017*).

## DISCUSSION

New technologies are needed to collect and characterize the distribution of microorganisms in natural and managed aquatic environments. We developed and implemented a remote-operated water sampler onboard an unmanned surface vehicle (USV) to collect samples at multiple depths and locations. Traditional water sampling has used human-powered vehicles such as kayaks (*Pietsch, Vinatzer & Schmale III, 2017*) and motorized boats (*Stangl, 2001*), but these missions are limited by direct human operation and line-of-sight operations. USVs have the potential to sample in remote areas, operate beyond the line-of-sight, and collect samples without any human intervention. Such operations are needed in aquatic environments to increase our understanding of changes in ecosystems and improve the accuracy of our long-term predictions.

The concentration of bacteria on both media types did not vary significantly between 5 and 50 cm, when controlling for sampling location. Thus, the bacteria appeared to be well-mixed between these two sampling depths. *Pietsch, Vinatzer & Schmale III (2017)* sampled water at the surface down to about 10 m, and observed no significant change in bacteria concentration with change in depth. Future work with the USV could probe deeper depths in the lake, to get a more accurate picture of the profile of microbes along the water column. It is possible that the difference in sampling depth (45 cm) was too small to reveal any real changes in bacterial concentrations. The maximum horizontal distance covered in this study was about 300 m in a small cove which could also be a factor. Other environmental factors, such as seasonal temperature variation and pH, are additional factors that could have an impact on bacterial distributions (*Lindströem, Kamst-Van Agterveld & Zwart, 2005*). Such seasonal and spatial changes are known to affect lake water chemistry, which in turn impacts microbial composition in lakes (*Parker et al., 2016*). We speculate that samples collected at different depths could indeed show differences in concentrations of bacteria during specific times of the year (i.e., during lake turnover) (*Wilhelm et al., 2014*).

There was a significant difference in concentrations on KBC across three sampling locations in the lake ($P = 0.027$), suggesting an uneven distribution of *Pseudomonas* across the locations sampled. KBC media is selective for bacteria in the genus *Pseudomonas*. Bacteria in this genus are rod shaped, Gram-negative, and flagellated. The flagella enable *Pseudomonas syringae* to have mobility. This mobility has been shown to be beneficial to the bacteria (*Lauffenburger, 1991*), and is effected by temperature (*Hockett, Burch & Lindow, 2013*). The difference in concentrations across locations may be related to this mobility, at least in part. The lack of significant change of concentration with location for the TSA media, could also be connected to this mobility (or lack thereof) for many bacteria. TSA is a general media type that is not selective for any particular bacteria, but is selective against fungal growth. Therefore, growth on this media includes both motile and non-motile bacteria. The difference in concentrations for the two locations (L1 and L2) may be related to geography; L1 was in a bend in the cove, offset slightly from the flow of the feeding creek from the south, and L3 was near the mouth of the cove opening up to the larger lake body (Fig. 1). L1 may be an area of high sedimentation caused by the

redirection of water flow from the cove into the lake due to this bend. Claytor Lake is a reservoir and has been reported to have a silt problem in the past (*Simmons, 2004*). The cove has a narrow channelized basin impacting the trophic classification of the location towards eutrophic (*Kimmel & Groeger, 1984*). Sediment, which can be high in organic content, is a point source for bacteria including pathogenic strains (*Kim et al., 2010*). For example, concentrations of fecal coliforms were orders of magnitude higher in sediments from an agricultural stream compared to the overlying water column (*Davies-Colley et al., 2004*). The disturbance and re-suspension of this sediment can lead to large releases of microorganisms (*Cinotto, 2005*), and should be considered in the design of future sampling campaigns.

There was no significant difference in the concentration of microbes across all days for KBC media. However, there was a significant difference in the concentration of microbes from day one to day three ($P = 0.038$) for TSA media when data from the second sampling mission on day three (final day) are not included in the statistical analysis. Previous studies have reported highly variable bacteria concentrations from day to day and location to location in the same day for Claytor Lake (*Pietsch, Vinatzer & Schmale III, 2017*). The undisturbed nature of the cove (limited human traffic with power boats and decreased wind interaction with the water surface due to tree cover) coupled with unique trophic characteristics, may contribute to the cove's decreased day to day variation in concentrations of bacteria (*Pietsch, Vinatzer & Schmale III, 2017*). Additional experiments including additional coves, depths, and seasons are needed to monitor any additional trends in microbial concentrations and potential feedback cycles.

About 1 percent (7/720) of the colonies from KBC media recovered across all four sampling missions were ice nucleation active (ice+) at temperatures warmer than $-10\,^\circ$C. Similar ratios of ice+ *Pseudomonas* and *Xanthomonas* were found in in simulated rain samples collected from a bridge 55 m above ground level (*Hanlon et al., 2017*) and from real rain events at multiple locations in Blacksburg, Virginia (*Failor et al., 2017*). *Pietsch, Vinatzer & Schmale III (2017)* found ∼6.9% of ice+ strains in the main water body of Claytor Lake across multiple seasons. This greater percentage of ice+ bacteria as compared to our study could be related to the open waters having exposure to greater wind/surface interactions and could suggest that larger open bodies of freshwater are preferential for some ice+ bacteria. There is an important feedback cycle with microorganisms and weather as they transport from aquatic environments into the atmosphere and back down to the ground (*Morris, Monteil & Berge, 2013*). These microorganisms are not only passengers along for a ride in the water cycle but may also serve as instigators for precipitation, a vital component of the water cycle. Some of these microorganisms pose health and economic risk while some serve important beneficial roles effecting the food web and climate such as *Pseudomonas syringae* which is both a plant pathogen and an ice nucleator that contributes to precipitation events across the globe (*Morris et al., 2014*). Understanding this complex interaction with aquatic environments and their inhabitants creates a rich field of study well suited to unmanned systems.

## CONCLUSIONS

A remote-operated water-sampling system was used onboard a USV to collect and characterize microbes in a freshwater lake in Virginia, USA. There was an uneven distribution of *Pseudomonas* across the locations sampled. There was no significant difference in concentrations of microbes across the two depths sampled, suggesting microorganisms were well-mixed between 5 and 50 cm below the surface of the water. About 1 percent (7/720) of the colonies recovered across all four sampling missions were ice nucleation active (ice+) at temperatures warmer than −10 °C. Our work extends traditional manned observations of aquatic environments to unmanned systems, and highlights the potential for USVs to understand the distribution and diversity of microbes within and above freshwater aquatic environments.

Tracking microorganisms such as *Pseudomonas syringae* throughout the entire water cycle could help to reveal a fundamental component of the water cycle and its role in both local and global weather and other important environmental processes (*Morris et al., 2008*). Such a study would require unmanned systems working together to sample in and above the water. Research conducted by autonomous coordinated systems (*Kolling et al., 2016*; *Vardy, 2016*) could reveal a wealth of information on transport patterns of microorganisms and associated environmental impacts. These studies could be comprised of surface, aerial and ground vehicles working as a heterogeneous swarm (*Szwaykowska, Romero & Schwartz, 2015*) to sample and characterize the environment with sophisticated meteorological sensors (e.g., windspeed, air and water temperature, solar radiation, etc.) and biological sensors (e.g., impingers (*Lin et al., 1999*) and optical particulate counters (*Lee et al., 2006*)). Such studies could also include laboratory measurements of CFU (*Pietsch, Vinatzer & Schmale III, 2017*), and flow cytometry measurements to characterize cell sizes and concentrations (*Buzatu et al., 2014*). Future work using these highly adaptive autonomous systems over diurnal and longer seasonal cycles could capture a more detailed picture of the role of these microorganisms in a range of environmental systems.

### Funding

This research was supported by the National Science Foundation (NSF) under Grant Numbers DEB-1241068 (Dimensions: Collaborative Research: Research on Airborne Ice-Nucleating Species (RAINS)), DGE-0966125 (IGERT: MultiScale Transport in Environmental and Physiological Systems (MultiSTEPS)), AGS 1520825 (Hazards SEES: Advanced Lagrangian Methods for Prediction, Mitigation and Response to Environmental Flow Hazards), and IIS-1637915 (NRI: Coordinated Detection and Tracking of Hazardous Agents with Aerial and Aquatic Robots to Inform Emergency Responders). There was no additional external funding received for this study. The funders had no role in study design, data collection and analysis, decision to publish, or preparation of the manuscript.

## Grant Disclosures

The following grant information was disclosed by the authors:
National Science Foundation (NSF): DEB-1241068, DGE-0966125, AGS-1520825, IIS-1637915.

## Competing Interests

The authors declare there are no competing interests.

## Author Contributions

- Craig Powers, Regina Hanlon and David G. Schmale III conceived and designed the experiments, performed the experiments, analyzed the data, contributed reagents/materials/analysis tools, wrote the paper, prepared figures and/or tables, reviewed drafts of the paper.

## Data Availability

The .stl (3D-printing) files are available in the Supplemental Files and at https://github.com/SchmaleLab/Schmale-Lab-3D-Printing-Files-Powers-et-al-PeerJ-2018.

## Supplemental Information

Supplemental information for this article can be found online at http://dx.doi.org/10.7717/peerj.4290#supplemental-information.

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
