# Peer review of "Remote collection of microorganisms at two depths in a freshwater lake using an unmanned surface vehicle (USV)"

_PeerJ, doi:10.7717/peerj.4290_

## Round 0.1 · original submission · Minor Revisions

Dear David,

I received two reviews of the manuscript on USV collection of water samples for microbial monitoring you submitted for consideration by Peer J. Both reviewers recommended acceptance following minor revisions. I have some editorial comments for your consideration too (below), if you choose to resubmit.

Michael

Line 14. The phrase “such as the bacterium Pseudomonas syringae” does not add much to the abstract. I understand the need to introduce that genus, but I am not sure this is the place to do that or the reason for sampling Pseudomonas was introduced adequately. Abstract and Introduction need some justification of why Pseudomonas were counted and assessed for ice nucleation.
Line 77. Suggest revision to “..in floodwaters generated by hurricanes and other extreme weather events.”
Line 145, 154. Insert space “47mm,” “150mg/L”
Line 259. Revise to “Sediment, which can be high in organic content, is a…For example, concentrations of fecal coliforms were orders of magnitude higher in sediments from an agricultural stream than the overlying water column..”
Line 273. Revise to “..may contribute to the cove’s..”
Line 350. Insert volume and pages
Line 354, 370 & 442. Cap journal titles (Applied and Environmental…Journal of Occupational…)
Line 382. Correct author list (Alex Huffman, P{\”..)
Line 390, 406. Italicize genus and species (Pseudomonas syringae)

Reviewer 1 ·

Basic reporting

The report is well written. Objectives, methods, discussion and conclusions are clear. Some minor editing is needed as described below.

Experimental design

Experiments utilized a custom designed sampling system to successfully collect water for microbial analyses at two different depths. Need some minor additional information to include approximated depth of the lake at the sample sites. Add this to Table 1 reference line 88.

end of line 133....How was the system sterilized between sample set collections

line 144....250ml.....spell out as it starts a sentence

line 162.....how many colonies......10 as the table indicates?....but several samples recovered less than 10cfu.......specify here and correct the table that states 10cfu sampled for these few samples

you should refer to the Pseudomona cfu's as presumptive Pseudomonas....as you ice nucleation experiments show that many were not.....this is common - colilert will indicate positive with certain species of Vibrios and Pseudomonads

Validity of the findings

Looks OK with addressing above comments. Check references of genus/species italics and standardize as one or a few use caps for many title words while the most use all lowercase

Reviewer 2 ·

Basic reporting

Language used throughout the manuscript is clear and well written. Only a few comments on this topic. There is a sentence in line 221 that may be correct, but sounds like it might has been a mistake during typing: “to collect samples and multiple depths and locations”, probably meant to be “to collect samples at multiple depths and locations”. Also lines 255 – 256, repetition of terms (location, due) should be avoided. In line 279, repetition of the preposition “in” should be corrected. Revision of sentence from line 297 to 304 is recommended.
Bibliography cited along the manuscript is extensive and recent, showing the authors are updated in their research field. There are however, corrections that have to be made to this section. Several papers cited in the References section are not mentioned in the text. Authors should include those citations where needed in the text or remove them from the References section. There is also one reference: Pietsch et al. (2007) (Line 44) which is not in the Reference list and it seems to be an error for what it should be Pietsch et al. (2017), which is widely cited along the manuscript. Although PeerJ Journal do not have a strict policy regarding the References section, they do have some guidelines. Authors should revise the entire section to comply to those guidelines, e.g. provide the full list of authors.
Article structure follows PeerJ Journal format. This paper contains 4 figures, which are helpful to the reader for the understanding, not only of the scientific results, but also for the development of the technology used to obtain them. They are presented in good resolution formats and well described. Table 1 and the supplemental table with all raw data are also well presented and clearly labeled.
Work presented in the manuscript is well organized and follows the objectives derived from the initial hypothesis stated by the authors in the introduction.

Experimental design

Research described in this manuscript fits within the Aims and Scope of PeerJ Journal. As mentioned before, it states a clear hypothesis with its specific objectives in the introduction that are fulfilled along the article. At first sight, it may seem a simple research work, with only 4 different samplings using culture-based methods, in a world where papers with big amounts of data are imposing. But the technology developed by the authors has a great potential in all kinds of research approaches and sometimes small research initiatives are needed in order to optimize that technology and/or methodology that could later be applied to more complex objectives. It is also known among the scientific community that environmental studies, although increasingly more frequent, haven’t provided the information needed to completely understand them. The use of USVs in environmental studies allows not only to reach possibilities beyond the human range, but also to those that could represent a risk to them.
Description of methodology allows replication of the experiments. Figure 3 and the ones included in the Supplemental files, plus the explanation in the Material and Methods section are elaborate, providing the capability of reproduction. Data in lines 186 – 192 from the Results section seem to repeat information already described in Material and Methods. Only new information related to the results should be included in this subheading.

Validity of the findings

Results obtained in the work presented add knowledge on the aquatic microbial community of a freshwater lake and the presence of ice nucleation active bacteria is a very important feature for the water cycle, as describe by the authors.
As considered by the authors in the discussion, collecting samples in different seasons as well as in different locations of the lake, would contribute to increase the understanding of how microbial communities change throughout the lake and are influenced by different environmental factors. Regarding this last topic, authors state in lines 90 - 91 of Material and Methods that “Though water temperature is an environmental variable, it was assume to be constant across the distances and times within and among samples collections in this study”, although no references or hypothesis that would back up this consideration are given. But later in the discussion section, lines 236 – 237, they acknowledge that “Other environmental factors, such as seasonal temperature variation and pH, are additional factors that could have an impact on bacterial distributions (Lindströem et al., 2005)”. Future efforts to complement this work should consider all environmental variables and authors may contemplate the possibility of including in the USV platform different sensors that would allow collecting that kind of information on site during the sampling process.
No conclusions are made in this manuscript, but speculations based on previous publications and the hypothesis of the authors. This is considered to be a wise decision regarding the small amount of samples collected and the statistical diversity of the results obtained.

---

## Round 0.2 · accepted · Accept

PeerJ has accepted your manuscript. Regards, Michael

Reviewer 1 ·

Basic reporting

no comment

Experimental design

no comment

Validity of the findings

no comment

Reviewer 2 ·

Basic reporting

No comment

Experimental design

No comment

Validity of the findings

No comment

Additional comments

Authors have addressed reviewers and editor's comments. Changes made and text added have improved comprehension of the manuscript. They have also adjusted bibliography format to the journal guidelines, and corrected those citations that were missing in either the main text of reference section.